# QAPyramid: Fine-grained Evaluation of Content Selection for Text Summarization

**Shiyue Zhang**[3][*][†]  **David Wan**[1][*]  **Arie Cattan**[2]  **Ayal Klein**[2]
**Ido Dagan**[2]  **Mohit Bansal**[1]
[1]UNC Chapel Hill  [2]Bar-Ilan University  [3]Bloomberg AI

## Abstract

How to properly conduct human evaluations for text summarization is a longstanding challenge. The Pyramid human evaluation protocol, which assesses content selection by breaking the reference summary into sub-units and verifying their presence in the system summary, has been widely adopted. However, it suffers from a lack of systematicity in the definition and granularity of the sub-units. We address these problems by proposing QAPyramid, which decomposes each reference summary into finer-grained question-answer (QA) pairs according to the QA-SRL framework. We collect QA-SRL annotations for reference summaries from CNN/DM and evaluate 10 summarization systems, resulting in 8.9K QA-level annotations. We show that, compared to Pyramid, QAPyramid provides more systematic and fine-grained content selection evaluation while maintaining high inter-annotator agreement without needing expert annotations. Furthermore, we propose metrics that automate the evaluation pipeline and achieve higher correlations with QAPyramid than other widely adopted metrics.[1]

## 1 Introduction

Human evaluation is considered the gold standard for benchmarking progress in text summarization (Kryscinski et al., 2019; Bhandari et al., 2020; Fabbri et al., 2021b; Celikyilmaz et al., 2021; Krishna et al., 2023), and provides the necessary training or evaluation signals for developing automatic metrics (Wei & Jia, 2021; Deutsch et al., 2021; Clark et al., 2021). However, there is no consensus on how human evaluation should be conducted. Flawed human evaluation protocol can undermine the reliability of any subsequent automatic evaluations or their outcomes. A key indicator of an unreliable human evaluation is the low inter-annotator agreement, which makes the evaluation results difficult to reproduce.

To make human evaluation more reliable, the Pyramid method (Nenkova & Passonneau, 2004) was introduced as a reference-based and decomposition-based human evaluation protocol. The reference defines what content should be selected for the summary, and decomposition reduces ambiguity when evaluating a long summary. Because of these, Pyramid is proved to be more reproducible compared to direct assessment (see more discussions in Section 2). In practice, Pyramid first decomposes the reference summary into Semantic Content Units (SCUs), defined as "semantically motivated, subsentential units," and then assesses whether each unit is *present* (semantically entailed) in the system-generated summary. The protocol has been continuously refined for efficiency and reproducibility (Shapira et al., 2019; Bhandari et al., 2020; Zhang & Bansal, 2021; Liu et al., 2023b). Despite the widespread recognition, we identify three significant issues with the underlying sub-unit decomposition step, on which the method is based.

First, the lack of a systematic definition for SCUs leads to ambiguity and inconsistency. Although Liu et al. (2023b) attempt to clarify the definition through the concept of Atomic

---

[*] Equal contribution.
[†] The work was conducted outside Bloomberg.
[1]Our data and code can be found in `https://github.com/ZhangShiyue/QAPyramid`.

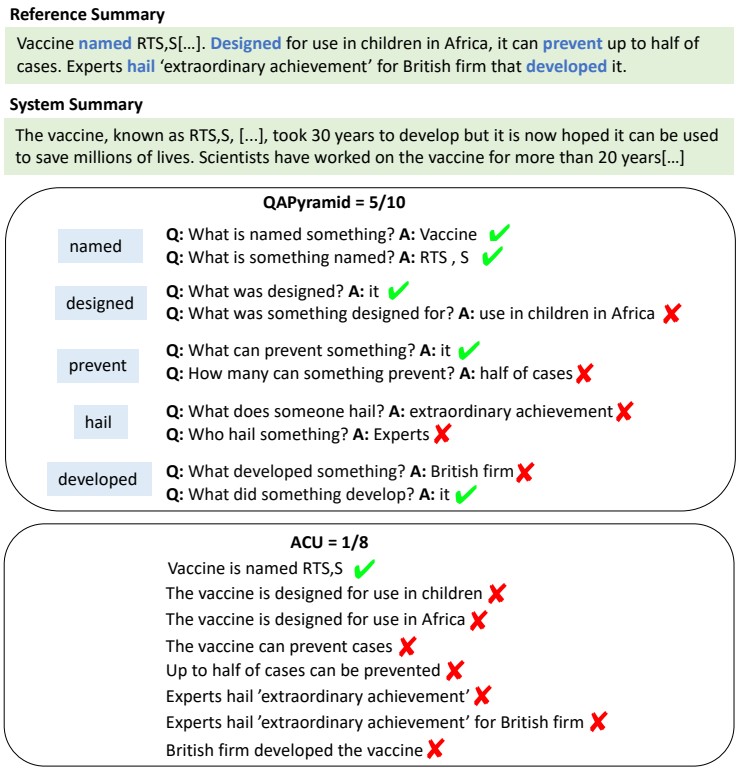

Figure 1: An illustration of our QAPyramid protocol and its comparison with the ACU protocol (Liu et al., 2023b). Compared to ACU, QAPyramid gives credit to finer-grained correctness like "What did something develop? it" ("it" refers to "the vaccine" in the reference summary). The system summary is generated by PEGASUS (Zhang et al., 2019). The example is from Liu et al. (2023b) and summaries are truncated due to space limit.

Content Units (ACUs), they acknowledge that "it can be impossible to provide a practical definition." Consequently, the content included in each unit varies among annotators, and the granularity of units is inconsistent, ultimately compromising its reproducibility. Second, the minimal SCU/ACU typically encompasses one predicate along with two or more arguments. If a system summary incorrectly represents even one of these arguments, it receives zero credit despite partial correctness. For instance, as illustrated in Figure 1, a system summary may state that "the vaccine took 30 years to develop" without mentioning who developed it. While this summary does not fully entail the semantics of "British firm developed the vaccine," it should still receive credit for correctly indicating that "the vaccine was developed." Therefore, a finer-grained representation is necessary to capture each predicate and its individual arguments separately, allowing for partial credit in evaluation. Third, due to the lack of systematicity, the Pyramid method relies on experts to extract and formulate units to ensure quality, making the protocol more costly and less scalable.

To address these problems, we introduce QAPyramid. Our method replaces the *SCU generation* step with a *QA generation* step that decomposes the reference summary into Question-Answer pairs (QAs) following the Question-Answer driven Semantic Role Labeling schema (QA-SRL He et al., 2015). Specifically, for each predicate (usually a verb) in the reference summary, annotators generate QA pairs, each corresponding to a "minimal" predicate-argument relation. In Figure 1, five predicates are identified from the reference summary and QAs are generated for each predicate. Then, annotators judge whether each QA is *present* (✔) or *not present* (✘) in the system summary and the final score is the number of present QAs over the total number of QAs. Figure 1 also illustrates the comparison between QAPyramid and ACU. QAPyramid demonstrates improvements over Pyramid-style methods for both systematicity and granularity. It not only provides a clear and consistent definition of content units as predicate-argument relations based on QA-SRL, but also de-

composes reference summaries into finer-grained units, leading to a more precise score. In addition, QA-SRL annotations are attainable in high quality via crowdsourcing (FitzGerald et al., 2018; Roit et al., 2020; Klein et al., 2020; Brook Weiss et al., 2021), which makes it more scalable compared to expert-annotated SCUs (Nenkova & Passonneau, 2004; Bhandari et al., 2020) or ACUs (Liu et al., 2023b).

We collect QAPyramid annotations on 500 English CNN/DM (Hermann et al., 2015) examples. First, we ask crowdsourced workers to write QA pairs for the reference summaries,[2] resulting in 8,652 QA pairs in total. On average, there are 17 QAs (compared to 11 ACUs) per reference summary. Then, across a subset of 50 examples[3] and 10 summarization systems, we collect 8,910 manual QA presence judgments. Our analysis reveals that in 21% of the cases, only a subset of the QA pairs for one predicate is present in the system summary, indicating the need for such finer-grained representation to capture partial correctness. Due to QA-SRL formalization, QAPyramid achieves high inter-annotator agreement and a high approval rate for generating QA pairs. And, despite requiring more granular judgments, QAPyramid attains an agreement level in detecting QA presence that is comparable to Pyramid-style human evaluation protocols.

Using the collected data, we explore approaches to automate QAPyramid. We automate QA generation and QA presence detection separately. We test off-the-shell fine-tuned models and few-shot LLM-based methods for both tasks. The former works the best for QA generation while the latter works the best for QA presence detection. With the best automation methods for both tasks, we develop two new metrics: a semi-automatic metric, *SemiAutoQAPyramid*, which automates only the QA presence detection (since QA generation only needs to be manually conducted *once* for any given dataset), and a fully automatic metric, *AutoQAPyramid*, which automates the entire pipeline. Compared to widely adopted metrics, these two new metrics achieve the highest correlations with the gold QAPyramid scores from human evaluations. This provides future research with finer-grained evaluation methods for text summarization at different levels of automation.

## 2   Background and Related Works

**Human Evaluation for Text Summarization.**   In many cases, human evaluation is conducted via *direct rating*, i.e., humans directly rate the quality of the summary (or rate for certain aspects, e.g., relevance (Fabbri et al., 2021b)). However, direct rating often suffers from subjectivity, low agreement, and thus non-reproducibility (Kiritchenko & Mohammad, 2017; Falke et al., 2019; Shapira et al., 2019). Two factors contribute to this issue: (1) what content is considered important and should be selected into the summary may vary from person to person, and (2) the summary is usually more than one sentence, and different annotations may put their focus on different parts of the summary. The canonical Pyramid (Nenkova & Passonneau, 2004; Shapira et al., 2019) method addressed these issues using reference summaries and decompositions. Recently, Liu et al. (2023b) revisited Pyramid and refined the definition of the sub-unit into atomic content units (ACU): "Elementary information units in the reference summaries, which no longer need to be further split for the purpose of reducing ambiguity in human evaluation." However, there are still no standard guidelines on how to write each unit and what content should be included in each unit. Hence, you may get different sets of units from different annotators, which undermines reproducibility. In response, we propose a new formalization using QA-SRL.

**QA-SRL.**   Semantic role labeling (SRL) is to discover the predicate-argument structure of a sentence, i.e., to determine "who does what to whom, when, and where," etc. Classic

---

[2]Some may challenge the quality of reference summaries in the CNN/DM dataset. However, this problem is orthogonal to our contribution. For any type of reference (good or bad), QAPyramid is more systematic, finer-grained, and more scalable than Pyramid-style protocols. Improving the quality of references is an orthogonal problem to be resolved.

[3]Due to budget limit, we only manually evaluate systems on a 50-example subset out of the 500. But the complete QAs of 500 examples are still valuable for evaluating systems on a larger set using our automated metrics in the future.

SRLs, e.g., FrameNet (Baker et al., 1998) and PropBank (Palmer et al., 2005), have rather complicated task definitions and require linguistic expertise to conduct annotations. He et al. (2015) introduced QA-SRL to present predicate-argument structure by question-answer (QA) pairs. Without predefining frames or semantic roles, the questions themselves define the set of possible roles. They showed that QA-SRL leads to high-quality SRL annotations and makes scalable data collection possible for annotators with little training and no linguistic expertise, which was later proved by the crowdsourced QA-SRL Bank 2.0 dataset (FitzGerald et al., 2018). The QA-SRL representation has since been utilized in various NLP tasks, demonstrating its versatility and effectiveness (Brook Weiss et al., 2021; Sultan & Shahaf, 2022; Caciularu et al., 2023; Cattan et al., 2024).

**Automatic Evaluation for Text Summarization.** Compared to human evaluations, automatic metrics are fast, cheap, and reproducible. However, how well they correlate with human judgment is always a concern. Over the years, many automatic evaluation metrics have been introduced. Some early metrics measure the n-gram overlap between system and reference summaries (Papineni et al., 2002; Lin, 2004; Banerjee & Lavie, 2005). To alleviate the rigidity of exact lexical match, metrics based on the similarity between embeddings were introduced (Zhao et al., 2019; Clark et al., 2019; Zhang* et al., 2020). There are also works automating the Pyramid protocol (Yang et al., 2016; Hirao et al., 2018; Gao et al., 2019; Zhang & Bansal, 2021; Liu et al., 2023c; Nawrath et al., 2024). They use Open IE, SRL, AMR, fine-tuned models, or LLMs to automate unit extraction and use NLI models to automate unit presence detection. Compared to these works, our AutoQAPyramid is advantageous because it automates a more systermatical, reproducible, and fine-grained human evaluation protocol, QAPyramid. Besides evaluating content selection, a separate line of research has been focused on faithfulness or factuality evaluation, i.e., checking if the source document(s) entail the summary (Cao et al., 2018; Maynez et al., 2020; Laban et al., 2022; Zha et al., 2023). The Pyramid type of methods have also been applied in this scenario (Chen et al., 2023; Min et al., 2023; Kamoi et al., 2023; Wan et al., 2024; Tang et al., 2024; Gunjal & Durrett, 2024). Some methods are based on question generation and answering (Wang et al., 2020; Durmus et al., 2020; Scialom et al., 2021; Fabbri et al., 2022). However, their QAs are not QA-SRL-based questions, and each QA contains more than one argument.

## 3 QAPyramid: Method and Dataset

### 3.1 QA Generation

For any given dataset, QA generation only needs to be done *once* for the reference summaries in its evaluation set. We choose CNN/DM (Hermann et al., 2015), one of the most popularly used text summarization datasets, as our test bed. We use a subset of 500 examples of the CNN/DM test set, the same set used by Liu et al. (2023b). For each reference summary, we first extract predicates automatically via AllenNLP SRL API (Gardner et al., 2018) which uses spaCy under the hood. On average, each reference contains 7.6 predicates.

Then, for each predicate, we ask one human annotator to write QA pairs. Figure 6 in the Appendix is our annotation UI on Amazon Mechanical Turk for writing QA pairs. Note that the previous QA-SRL annotation (FitzGerald et al., 2018) requires annotators to follow predefined automata, which sometimes results in non-natural questions. Here we opt for a less confined approach to allow annotators to write questions following our instructions (instructions can be seen in Figure 6). We show one sentence at a time and highlight one predicate in the sentence. The annotator is asked to write at most 5 questions and, for each question, fill in at most 3 answers. To recruit high-quality annotators for this task, we initially picked 4 examples as qualification tasks. Workers were qualified only if they correctly wrote QA pairs for all 4 tasks. Eventually, we recruited 31 annotators.

For each of the collected QA pairs, following FitzGerald et al. (2018), we ask two other annotators to verify it. This step validates if QAs are correctly written based on our instructions. Figure 7 in the Appendix is the UI for verifying QA pairs. The annotator is asked to first judge whether the question is valid; if valid, they need to write the answer to the question. Similar to QA generation, we used 4 qualification tasks to recruit 30 annotators. In addition,

we conduct cross-checks between QA generation and verification. Annotators for writing QA pairs need to get more than 85% accuracy in the verification step to maintain being qualified. Annotators for verifying QA pairs need to agree with their peer annotators for more than 85% to maintain being qualified. Annotators who write QA pairs are compensated at a rate of $0.32 per HIT, and annotators who verify QA pairs are paid at a rate of $0.17 per HIT, which makes an hourly payment of around $10. We find a high inter-annotator agreement: in 90.7% of cases, two annotators agree with each other, and in 89.7% of cases, the question is labeled as valid by both annotators. These high agreement and approval rates indicate that QA generation, following the QA-SRL formalization, is systematically standardized, making it verifiable and reproducible. In the end, we only keep QA pairs that are verified to be valid by both annotators. If a predicate has fewer than 2 QA pairs, we redo QA generation and verification.

Eventually, we collected a total of 8,652 QA pairs, averaging 17 QAs per reference summary. On the same dataset, there are on average 11 ACUs (Liu et al., 2023b) per reference summary, which confirms the finer granularity of our evaluation protocol.

## 3.2 QA Presence Detection

After we obtained all the QA pairs, the next step is to conduct system evaluations. For any system summary, we ask humans to judge whether each QA is *present* (✔) or *not present* (✘) in the system summary. Figure 8 in the Appendix shows the annotation UIs of this task. We instruct annotators to judge a QA pair as being present if its meaning is covered or implied by the system summary, or can be inferred from the system summary, while the exact format, expression, or wording can be different. Note that the reference summary is also provided in this task for annotators to ground each QA and infer any necessary coreference information, e.g., in Figure 1, we can easily know "it" refers to "the vaccine" based on the reference. Same as QA generation and verification tasks, we used 4 HITs as qualification tasks, and only employed annotators who got more than 90% accuracy. We recruited 27 annotators. To reduce workload, we only display the QA pairs for one predicate (usually 2 or 3 QAs) in one HIT and pay $0.2/HIT, which makes a $10 hourly rate.

Due to budget constraints, we randomly subsampled 50 examples from the 500 CNN/DM test examples to conduct system evaluation and evaluated 10 systems, including 5 models that are fine-tuned on CNN/DM: BART (Lewis et al., 2020), PEGASUS (Zhang et al., 2019), BRIO (Liu et al., 2022), BRIO-Ext, and MatchSum (Zhong et al., 2020), and 5 LLMs that generate summaries via 1-shot learning: Llama-3-8B-Instruct (Llama Team et al., 2024), Llama-3-70B-Instruct, Mixtral-8×7B-Instruct (Jiang et al., 2024), Mixtral-8×22B-Instruct, and GPT-4 (*gpt-4-0125*). In total, we collected 8,910 QA presence judgments. Each judgment was provided independently by three annotators, and the final decision was determined by a majority vote. The average inter-annotator agreement (IAA) is 0.74 (Krippendorff's alpha). For reference, Liu et al. (2023b) reported an IAA of 0.75, Zhang & Bansal (2021) reported 0.73, and Bhandari et al. (2020) reported 0.66.

## 3.3 Summary Scoring

Given human annotations, we now can calculate the QAPyramid scores. For any given system summary $s_i$, assume its reference summary $r_i$ has $K_i$ QA pairs $\{\text{QA}_{ij}\}_{j=1}^{K_i}$, then QAPyramid is defined as the number of present QA pairs in $s_i$ divided by $K_i$:

$$\text{QAPyramid}_i = \frac{\sum_{j=1}^{K_i} \text{Presence}(\text{QA}_{ij}, s_i)}{K_i},$$

where $\text{Presence}(\text{QA}_{ij}, s_i) = 1$ if $\text{QA}_{ij}$ is present in $s_i$, and 0 otherwise.

This metric is essentially a *recall*, i.e., a longer summary with more information usually gets a higher score. In an extreme case, when the summary is the same as the source document, it receives a perfect score. To combat this issue, Liu et al. (2023b) introduced a normalized ACU score, $n$ACU, by multiplying the original ACU score by a *length penalty*, i.e., $n$ACU

| | | QAPyramid | $n$QAPyramid | ACU | $n$ACU | R2-R | R2-F1 | Length |
|---|---|---|---|---|---|---|---|---|
| *FT* | BART | 0.51 | 0.48 | 0.37 | 0.29 | 0.23 | 0.20 | 69.54 |
| | PEGASUS | 0.46 | 0.44 | 0.35 | 0.30 | 0.23 | 0.21 | 63.46 |
| | BRIO | **0.56** | **0.53** | **0.43** | **0.35** | **0.27** | **0.24** | 66.46 |
| | BRIO-Ext | **0.56** | 0.52 | 0.41 | 0.32 | 0.25 | 0.22 | 69.86 |
| | MatchSum | 0.51 | 0.46 | 0.41 | 0.31 | 0.25 | 0.21 | 74.06 |
| *LLM* | Llama-3-8B-Instruct | 0.54 | 0.40 | - | - | 0.23 | 0.14 | 272.94 |
| | Llama-3-70B-Instruct | 0.53 | 0.47 | - | - | 0.18 | 0.14 | 82.88 |
| | Mixtral-8×7B-Instruct | 0.48 | 0.45 | - | - | 0.19 | 0.17 | 67.44 |
| | Mixtral-8×22B-Instruct | 0.48 | 0.44 | - | - | 0.23 | 0.18 | 74.58 |
| | GPT4 | 0.55 | 0.46 | - | - | 0.19 | 0.13 | 102.74 |
| | Reference | 1.00 | 1.00 | 1.00 | 1.00 | 1.00 | 1.00 | 57.74 |

Table 1: The evaluation results of different summarization systems on 50 CNN/DM testing examples. QAPyramid and $n$QAPyramid are unnormalized and normalized QAPyramid scores. ACU and $n$ACU are unnormalized and normalized ACU scores (Liu et al., 2023b). R2-R and R2-F1 are ROUGE-2 recall and f1 scores (Lin, 2004). Length is the average summary length in tokens. Table 6 contains systems scores of many other metrics. *FT* means the models fine-tuned on CNN/DM training set, and *LLM* means the 1-shot LLM-based models

$= p_i^l * \text{ACU}$, where $p_i^l = e^{\min(0, \frac{1 - \frac{|s_i|}{|r_i|}}{\alpha})}$ and $|s_i|, |r_i|$ are the length (i.e., number of words) of the system and reference summary. Essentially, system summaries that are longer than the reference summary get discounted scores. In practice, $\alpha$ is set by de-correlating $n$ACU with summary length.

However, this length penalty, especially how $\alpha$ is being set, assumes the unnormalized score is positively correlated with summary length, i.e., longer summaries tend to get higher scores. This is usually true for fine-tuned models. But for non-finetuned LLMs, they sometimes suffer from the *degeneration* problem (Holtzman et al., 2020) – getting stuck into a repetition loop and generating a sentence or a sub-sentence repeatedly, see a degenerated summary produced by Llama-3-8B-Instruct in Figure 2. In this case, the extending length does not provide more information and thus does not lead to a higher score. Though this extending length also needs to be penalized, it has unique behavior and needs to be dealt with separately. Therefore, we introduce a novel *repetition penalty* $p_i^r = 1 - rp_i$, where $rp_i$ is the repetition rate of $s_i$ meaning what percentage of the summary is repetition (please refer

to Figure 2 in Appendix A.2). Then we change the length penalty to $p_i^l = e^{\min(0, \frac{1 - \frac{|s_i| * p_i^r}{|r_i|}}{\alpha})}$, where $|s_i| * p_i^r$ is the *effective* summary length – the length of non-repetitive text. We set $\alpha$ by de-correlating $p_i^l * \text{QAPyramid}$ with *effective* summary length. Using our collected data, the Pearson correlation between *effective* summary length and QAPyramid is 0.27. After setting $\alpha = 6$, the correlation between *effective* summary length and $p_i^l * \text{QAPyramid}$ reduced to -0.01 (see details in Table 5 in Appendix A.2).

The normalized QAPyramid is defined as:

$$n\text{QAPyramid} = p_i^r * p_i^l * \text{QAPyramid}.$$

Intuitively, $p_i^r$ penalizes long summary that has a lot of repetitions, and $p_i^l$ penalizes long summary that includes a lot of information from source.

### 3.4 Result Analysis

Table 1 shows the metric scores of 10 summarization systems on the 50-example subset from the CNN/DM test set (and Table 6 in the Appendix contains systems scores of many other automatic or semi-automatic metrics). The ACU and $n$ACU scores are computed based on the raw annotation data released by Liu et al. (2023b).[4] First, a consistent trend across different metrics is that BRIO and BRIO-Ext are the two best-performing models, likely

---

[4]https://huggingface.co/datasets/Salesforce/rose

because they are carefully fine-tuned on the CNN/DM training set. Second, 1-shot LLMs consistently obtain worse ROUGE-2 recall or F1 scores than fine-tuned models. However, their QAPyramid and $n$QAPyramid scores are comparable to those of fine-tuned models. This demonstrates that, compared to metrics based on lexical overlaps, our QAPyramid method captures more semantic similarities and thus alleviates the rigidity caused by reference-based evaluations with a single reference. This finding is also consistent with previous observations that zero-shot or few-shot LLMs are preferred by humans despite their low ROUGE scores (Goyal et al., 2023). Third, we observe that QAPyramid scores are overall higher than ACU scores. One hypothesis is that QAs are finer-grained than ACUs, so they give credit to finer-grained alignments between the system and reference summaries. To support this hypothesis, we find that in 21% of cases, a predicate only has a subset of its QA pairs present in the system summary, i.e., if the judgment is at the ACU level, it would miss some partial correctness. Next, we manually examined some examples and we find that, besides QA pairs being finer-grained, two other factors also contribute to higher QAPyramid scores: (1) our annotation guideline led to more lenient judgments of "being present" based more on semantics than lexicon, and (2) the predicate-centered nature of QA-SRL may cause one piece of information to be credited multiple times. See Appendix C for two examples.

## 4 QAPyramid Automation

### 4.1 QA Generation Automation

For the QA generation task, the input is one reference summary and one predicate (verb) within the reference, and the gold output is the human-written QA pairs for this predicate. Using our collected data, we get a set of 3,782 examples. Since we do not plan to train a QA generation model, we use all of them as our test set.

We explore two types of models to conduct this task automatically. First, we consider prompting large language models (LLMs). We use open-sourced LLMs, Llama-3.1-8B-Instruct and Llama-3.1-70B-Instruct (Llama Team et al., 2024), as well as a proprietary model, GPT-4o (OpenAI, 2024). We test with 0, 1, 3, and 5 in-context examples[5] randomly sampled from the set. To prevent answer leakage, we ensure that the in-context examples do not contain the same reference summary as the test example. LLM prompts are shown in Figure 7. Second, we use a fine-tuned model, QASem (Klein et al., 2022), which was jointly trained on the QASRL (FitzGerald et al., 2018) and QANom (Klein et al., 2020) datasets based on T5 models. The parser takes as input a sentence and a predicate and generates a set of QA pairs. In our experiments, we use an improved version of the parser (Cattan et al., 2024), which leverages Flan-T5-Large and Flan-T5-XL (Shen et al., 2023).[6]

To evaluate how similar the generated QAs are to those written by humans, we assess the similarity between the generated and gold QA pairs using ROUGE-L (Lin, 2004), BERTScore (Zhang* et al., 2020), and a previously developed metric from QASRL (Roit et al., 2020) and QANom (Klein et al., 2020) tasks, Unlabeled Argument Detection (UA). UA calculates token overlap between the generated and gold answers.[7]

We report the results in Table 2. For the LLM-based models, we observe a clear trend: having more in-context examples consistently improves performance. For example, the UA metric is more than doubled from zero-shot to five-shot settings. Within the Llama-3.1 models, the larger model (70B) outperforms the smaller model (8B) on all metrics and in all in-context settings, except for the zero-shot setting with BERTScore. Interestingly, the performance of Llama-3.1-70B-Instruct is quite comparable to that of GPT-4o, achieving similar BERTScore and ROUGE-L scores. This demonstrates the promise of using open-source models for this QA generation task. Finally, we observe that the two fine-tuned models achieve the highest

---

[5]We do not find any significant gains using more than 5 in-context examples.

[6]https://github.com/plroit/qasem_parser

[7]We exclude labeled argument detection, the metric on question equivalence because it relies on a rule-based parser that works only on questions that follow predefined automata and is therefore not suitable for our "free-form" questions, as mentioned in Section 3.1.

| QA Generation | | | | | QA Presence | | | |
|---|---|---|---|---|---|---|---|---|
| Method | | RL | BS | UA | Method | | QA F1 | Stmt F1 |
| QASem (flan-t5-large) | | 78.9 | **94.5** | **99.9** | DeBERTa v3 | | 81.6 | 79.3 |
| QASem (flan-t5-xl) | | **79.0** | **94.5** | **99.9** | MiniCheck | | 78.8 | 80.1 |
| | | | | | AlignScore | | 77.3 | 76.9 |
| Llama-3.1-8B-Inst | 0shot | 36.9 | 87.9 | 24.6 | Llama-3.1-8B-Inst | 0shot | 53.8 | 69.6 |
| | 1shot | 49.4 | 90.0 | 47.7 | | 1shot | 79.9 | 77.4 |
| | 3shot | 56.9 | 91.2 | 57.2 | | 3shot | 84.8 | 82.3 |
| | 5shot | 61.2 | 91.9 | 60.8 | | 5shot | 81.5 | 78.3 |
| Llama-3.1-70B-Inst | 0shot | 41.3 | 87.1 | 32.1 | Llama-3.1-70B-Inst | 0shot | 81.5 | 80.4 |
| | 1shot | 61.3 | 91.3 | 66.1 | | 1shot | 84.9 | 80.6 |
| | 3shot | 69.4 | 92.8 | 72.7 | | 3shot | 84.8 | 82.3 |
| | 5shot | 72.0 | 93.3 | 74.6 | | 5shot | 84.7 | 83.0 |
| GPT-4o | 0shot | 45.9 | 88.9 | 41.5 | GPT-4o | 0shot | 78.8 | 81.8 |
| | 1shot | 62.0 | 91.4 | 71.0 | | 1shot | 82.9 | 83.5 |
| | 3shot | 69.4 | 92.7 | 76.7 | | 3shot | 84.7 | 83.7 |
| | 5shot | 72.2 | 93.3 | 78.2 | | 5shot | **85.0** | **84.2** |

Table 2: Automatic QA generation and presence detection performance. RL, BS, and UA refer to ROUGE-L, BERTScore, and unlabeled argument detection, respectively. For QA presence detection, we report micro F1 scores in two scenarios where we take the QA pair as is or transform it into a statement (Stmt).

scores, which is expected since they have been trained to generate QAs of similar styles. Thus, we recommend using fine-tuned QASem models to automate the QA generation step.

## 4.2 QA Presence Detection Automation

For the QA presence detection task, the input is one system summary and one QA pair from the reference summary, and the output is a binary judgment of whether the QA can be inferred from the system summary.[8] The gold output label is the majority vote among annotators, resulting in 8,910 examples for this task. Similar to QA generation, we use all examples as our test set.

Similar to QA generation, we explore LLM-based models with different numbers of in-context examples, excluding examples that use the same reference summary. Since detecting the presence of QAs is a typical natural language inference (NLI) task, we explore an off-the-shelf pretrained NLI model, DeBERTa v3[9] (He et al., 2023). Additionally, we test two NLI-based faithfulness evaluation metrics for summarization: MiniCheck (Tang et al., 2024) and AlignScore (Zha et al., 2023). They were found to highly correlate with human judgments in whether a system summary is entailed by the document. In typical entailment tasks, the hypothesis is usually a statement rather than a QA. Therefore, we also explore transforming the QAs into *statements* using GPT-4o. Finally, we report the micro F1 score to assess how well the predicted labels match the gold labels.

The results are shown in Table 2. We observe that DeBERTa achieves performance comparable to zero-shot GPT-4o, demonstrating the efficacy of applying NLI models pre-finetuned on many NLI tasks. Compared to specialized models for assessing faithfulness, DeBERTa achieves higher correlations. For LLM-based models, we find that providing more in-context examples generally improves performance. In particular, we observe a large improvement from zero-shot to one-shot; however, the improvement seems to plateau between three-shot and five-shot. Similar to the results of QA generation automation, we also observe that the

---

[8]In our initial experiment, we also attempted to include the reference summary for additional context, but this resulted in worse performance. We leave further exploration of this approach to future work.

[9]We use the model that has been fine-tuned on many tasks for strong classification performance: sileod/deberta-v3-base-tasksource-nli.

|  | Metric | FT | | LLM | | All | |
|---|---|---|---|---|---|---|---|
|  |  | System | Summary | System | Summary | System | Summary |
| *Manual* | ACU | 0.800 | 0.435 | - | - | - | - |
| *Semi-automatic* | SemiAutoACU | 0.800 | 0.508 | 0.200 | 0.350 | 0.556 | 0.476 |
|  | Lite$^2$Pyramid w. ACU | 0.800 | 0.503 | 0.200 | 0.501 | 0.467 | 0.564 |
|  | SemiAutoQAPyramid | **1.000** | **0.603** | **0.800** | **0.537** | **0.956** | **0.630** |
| *Fully Automatic* | ROUGE-1 | 0.800 | 0.459 | 0.400 | 0.437 | 0.556 | 0.500 |
|  | ROUGE-2 | 0.600 | 0.445 | -0.200 | 0.378 | 0.156 | 0.398 |
|  | ROUGE-L | -0.200 | 0.351 | 0.200 | 0.354 | 0.156 | 0.380 |
|  | METEOR | 0.800 | 0.448 | -0.200 | 0.293 | 0.200 | 0.405 |
|  | CHRF | 0.800 | 0.461 | -0.200 | 0.293 | 0.289 | 0.423 |
|  | BERTScore | 0.800 | 0.471 | 0.200 | 0.384 | 0.289 | 0.449 |
|  | BARTScore | 0.800 | 0.470 | 0.200 | 0.459 | 0.467 | 0.507 |
|  | GEval | 0.600 | 0.289 | 0.000 | 0.279 | 0.422 | 0.366 |
|  | AutoACU | 0.600 | 0.444 | 0.200 | 0.394 | 0.511 | 0.476 |
|  | Lite$^3$Pyramid | **1.000** | 0.460 | 0.600 | 0.467 | 0.689 | **0.558** |
|  | AutoQAPyramid | 0.600 | **0.508** | **0.800** | **0.510** | **0.733** | 0.549 |

Table 3: System and summary level Kendall's correlations between the metric scores and gold QAPyramid scores. We bold the best metrics for semi-automatic or fully automatic settings respectively. *FT* means the 5 fine-tuned models, *LLM* means the other 5 LLM-based models, and *All* means all 10 systems.

larger model (70B) outperforms its smaller counterpart (8B) and that open-source models are competitive with the proprietary model, especially in zero-shot and one-shot settings. Surprisingly, the F1 scores we obtained when we used QA as is are mostly higher than or comparable to when we converted the QA into a statement. We believe this may be due to errors introduced by the QA-to-statement transformation. Overall, the best metric here is using GPT-4o with 5-shot examples, followed by Llama-3.1-70B-Inst 1 shot that only performs worse by 0.1%.

### 4.3 Meta Evaluation of Automated Metrics

Finally, equipped with the best automation methods, i.e., QASem with flan-t5-xl for QA generation and GPT-4o with 5-shot prompting for QA presence detection, we can assemble both a semi-automatic and a fully automatic metric. For the semi-automatic metric, *SemiAutoQAPyramid*, we automate only the QA presence detection part, allowing any new system to be evaluated on the same test set that already have human-written QAs. For the fully automatic metric, *AutoQAPyramid*, we further automate QA generation, enabling QAPyramid to be automatically applied to any new test set and/or any new system.

Since we believe QAPyramid provides more reliable and accurate human evaluation signals, we use it to benchmark automated metrics. We test the correlation between QAPyramid and another Pyramid-style human evaluation protocol, ACU (Liu et al., 2023b), and its automated version, AutoACU (Liu et al., 2023c) (we use A2CU). Although Liu et al. (2023c) did not explore the semi-automatic option, we test SemiAutoACU metric using their human-written atomic units and their trained unit presence checking model. We also include metrics introduced by Zhang & Bansal (2021) that semi-automate or fully automate the Pyramid method, namely Lite$^2$Pyramid and Lite$^3$Pyramid, respectively. For Lite$^2$Pyramid, we utilize the gold units from ACU (because SCUs are not available for the 50 examples in our dataset) while employing Zhang & Bansal (2021) trained presence detection model.

Lastly, we include various widely adopted summarization evaluation metrics: ROUGE (Lin, 2004), METEOR (Banerjee & Lavie, 2005), CHRF (Popović, 2015), BERTScore (Zhang* et al., 2020), BARTScore (Yuan et al., 2021), and a variant of GEval (Liu et al., 2023a)[10], a GPT-4-based metric that predicts a numerical score. Since QAPyramid is recall-focused, we use the recall version of these metrics when available.

Following previous works (Fabbri et al., 2021b; Liu et al., 2023b), we use system-level and summary-level Kendall's tau correlations as our meta-evaluation metrics. The results are

---

[10]We adapted the prompt from Liu et al. (2023b) for this task.

presented in Table 3. We report the results separately and collectively for the 5 fine-tuned (FT) models and the 5 LLM-based models. First, for both ACU and QAPyramid, their semi-automatic metrics outperform their fully automatic counterparts. Surprisingly, Lite$^3$Pyramid works better than Lite$^2$Pyramid with ACU perhaps due to the mismatch between ACUs and the original SCUs used by Lite$^2$Pyramid. Second, we find that AutoACU or Lite$^3$Pyramid, which uses ACUs or SCUs, has lower correlations than AutoQAPyramid. This is due to the mismatch between their original Pyramid-style human evaluations and QAPyramid. Nonetheless, we believe QAPyramid is a more reliable protocol and it is desired to best correlate with it. Lite$^3$Pyramid correlates surprisingly well with QAPyramid probably because their automated SCUs (called *STUs*) are extracted based on SRL. Overall, our SemiAutoPyramid and AutoQAPyramid metrics show higher correlations with QAPyramid than other metrics. This demonstrates that they effectively automate QAPyramid, thus providing finer-grained automatic evaluation methods for text summarization.

## 5 Conclusion

To summarize, the contributions of our work are three-fold. First, we introduce QAPyramid, an enhanced human evaluation protocol for text summarization that improves the systematicity, granularity, reproducibility, and scalability of reference-based and decomposition-based human evaluations. Second, we extensively collect 8.9K annotations following our QAPyramid protocol, which can be used to benchmark automatic metrics and summarization systems in the future. Third, we introduce semi-automatic and fully automatic metrics that partially or entirely automate QAPyramid, and they show the highest correlations with QAPyramid compared to other widely adopted summarization metrics. We hope that QAPyramid and its automation can be adopted by future work for benchmarking text summarization and possibly other language generation tasks.

## 6 Limitations

First, our evaluation is confined to the English CNN/DM news summarization dataset; the generalizability of our findings to other languages and domains remains to be explored.

Second, as a Pyramid-style metric, QAPyramid is strictly reference-based. This reliance on human-authored references can be a constraint, as they may be unavailable or biased if only a single reference is provided. This limitation is evident in our experiments on the reference-free SummEval benchmark (Appendix A.1), where we observe weak correlations. The discrepancy arises because our metric is tightly grounded in the reference summary, whereas human ratings often permit greater flexibility in content selection.

Finally, our semantic decomposition could be more comprehensive. We currently use only QA-SRL (He et al., 2015; FitzGerald et al., 2018) to generate question-answer (QA) pairs, but future work could incorporate methods for nominalizations (Klein et al., 2020) or discourse relations (Pyatkin et al., 2020) for a more thorough analysis. Relatedly, our metric weights all QA pairs equally, which may allow numerous minor arguments to overshadow more salient information. This could be addressed by exploring predicate-level aggregation, where scores are averaged within a predicate before being combined.

## Acknowledgments

We thank the anonymous reviewers for their helpful comments, and we also thank Paul Roit, Ori Shapira, and Ori Ernst for the helpful discussions. This work was supported by NSF-CAREER Award 1846185, NSF-AI Engage Institute DRL-2112635, DARPA Machine Commonsense (MCS) Grant N66001-19-2-4031, Microsoft Accelerate Foundation Models Research (AFMR) grant program, a Google PhD Fellowship, the Israel Science Foundation (grant no. 2827/21), by a grant from the Israeli Planning and Budgeting Committee (PBC), and the PBC fellowship for outstanding PhD candidates in data science. The views contained in this article are those of the authors and not of the funding agency.

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

# A Complementary Results

## A.1 Correlations with Other Annotation Benchmarks

We evaluate *AutoQAPyramid* on the SummEval benchmark (Fabbri et al., 2021a). Since SummEval provides multiple reference summaries, we test both single-reference and multi-reference variants of our metric, correlating their outputs against the benchmark's human

|  | System-level | | | Summary-level | | |
|---|---|---|---|---|---|---|
|  | r | $\rho$ | $\tau$ | r | $\rho$ | $\tau$ |
| *AutoQAPyramid* single-ref | **0.27** | 0.02 | -0.03 | 0.06 | 0.05 | 0.04 |
| *AutoQAPyramid* multi-ref | 0.26 | **0.12** | **0.07** | **0.10** | **0.10** | **0.07** |

Table 4: Pearson (r), Spearman ($\rho$), and Kendall ($\tau$) correlations of *AutoQAPyramid* on SummEval.

relevance scores. The multi-reference variants computes the score by taking the average across triplets extracted from all references. The results, presented in Table 4, show that the multi-reference variant outperforms the single-reference one in five out of six cases.

Despite this, the overall correlations with human judgments are lower than expected. We hypothesize this is due to a protocol mismatch: our metric is reference-based, but SummEval's annotations are reference-free (annotators were not shown a reference summary). Such a discrepancy is known to depress correlations, with Liu et al. (2023b) finding a system-level Pearson's correlation of -0.25 between reference-based and reference-free human judgments. Indeed, when we compare our reference-based metric with another reference-based metric, ROUGE-1, we observe a strong Pearson correlation of 0.73. This confirms our metric is performing correctly but is misaligned with SummEval's reference-free setup.

This finding highlights a critical gap in existing resources and validates the need for our newly collected human annotations, which provide a benchmark specifically designed to assess fine-grained, reference-aware relevance between summaries.

### A.2 Length and Repetition Penalty

To determine the repetition rate $rp_i$ of a summary, we first identify the span that is repeated consecutively for more than 3 times. In the example of Figure 2, "Gwyneth Paltrow and Chris Martin are seen together with their children in a family photo ." is the span. Then we identify the text $t_i$ where the span is being repeated; in this case, it is the length of text in red. The repetition rate $rp_i$ is then defined by $\frac{|t_i|}{|s_i|}$, i.e., the length of repetitive text over the summary length.

---

**Reference**: Gwyneth Paltrow filed for divorce from Chris Martin a year after splitting. Our expert says there's much to applaud in their civilised separation. But says aspects of their 'conscious uncoupling' should be avoided.

**Summary (generated by Llama-3-8b-Instruct)**: Gwyneth Paltrow and Chris Martin's amicable split has been praised for being a refreshing change from the usual celebrity break-ups . Relationship expert Tracey Cox says that the couple's approach to their split can be learned from, including putting their children's needs first, not criticising each other's new partners, and focusing on what they liked about each other at the start . However, Cox also warns that not all couples can follow in their footsteps, and that some aspects of their approach, such as not having a separation period, may not be suitable for everyone. Gwyneth Paltrow and Chris Martin are seen together with their children in a family photo . Chris Martin and Gwyneth Paltrow are seen together with their children in a family photo . Gwyneth Paltrow and Chris Martin are seen together with their children in a family photo . Gwyneth Paltrow and Chris Martin are seen together with their children in a family photo . Gwyneth Paltrow and Chris Martin are seen together with their children in a family photo. [...]

---

Figure 2: An example of a degenerated summary, and repetitive text is marked by red.

To de-correlate $p_i^l * \text{QAPyramid}$ with the *effective* summary length, we enumerate $\alpha$ from 1 to 10 and pick the value when the correlation is the lowest, which is when $\alpha = 6$.

### A.3 System Scores

We show the system scores in Table 6.

| $\alpha$ | 1 | 2 | 3 | 4 | 5 | 6 | 7 | 8 | 9 | 10 |
|---|---|---|---|---|---|---|---|---|---|---|
| pearsonr | -0.46 | -0.27 | -0.16 | -0.09 | -0.04 | -0.01 | 0.02 | 0.04 | 0.06 | 0.07 |

Table 5: The pearson correlation between $p_i^l * \text{QAPyramid}$ and *effective* summary length with different $\alpha$s.

| | BART | PEGASUS | BRIO | BRIO-Ext | MatchSum | Llama-3-8B-Inst. | Llama-3-70B-Ins. | Mixtral-8×7B-Inst. | Mixtral-8×22B-Inst. | GPT-4 |
|---|---|---|---|---|---|---|---|---|---|---|
| ACU | 0.37 | 0.35 | **0.43** | 0.41 | 0.41 | | | | | |
| nACU | 0.29 | 0.3 | **0.35** | 0.32 | 0.31 | | | | | |
| QAPyramid | 0.51 | 0.46 | **0.56** | 0.55 | 0.5 | 0.54 | 0.53 | 0.48 | 0.48 | 0.55 |
| nQAPyramid | 0.47 | 0.44 | **0.53** | 0.52 | 0.46 | 0.4 | 0.47 | 0.45 | 0.44 | 0.46 |
| SemiAutoACU | 0.36 | 0.35 | **0.43** | 0.41 | 0.38 | 0.38 | 0.35 | 0.31 | 0.37 | 0.37 |
| Lite²Pyramid w. ACU | 0.45 | 0.43 | **0.5** | 0.49 | 0.46 | 0.44 | 0.44 | 0.4 | 0.44 | **0.5** |
| SemiAutoQAPyramid | 0.47 | 0.41 | **0.52** | 0.51 | 0.46 | 0.5 | 0.48 | 0.45 | 0.46 | 0.51 |
| nSemiAutoQAPyramid | 0.44 | 0.39 | **0.49** | 0.47 | 0.43 | 0.36 | 0.44 | 0.42 | 0.42 | 0.43 |
| ROUGE-1-R | 0.49 | 0.48 | 0.54 | 0.51 | 0.51 | **0.55** | 0.5 | 0.46 | 0.5 | 0.52 |
| ROUGE-1-F1 | 0.42 | 0.44 | **0.47** | 0.44 | 0.42 | 0.34 | 0.39 | 0.4 | 0.41 | 0.36 |
| ROUGE-2-R | 0.23 | 0.23 | **0.27** | 0.25 | 0.25 | 0.23 | 0.18 | 0.19 | 0.23 | 0.19 |
| ROUGE-2-F1 | 0.2 | 0.21 | **0.24** | 0.22 | 0.21 | 0.14 | 0.14 | 0.17 | 0.18 | 0.13 |
| ROUGE-L-R | 0.34 | 0.34 | **0.37** | 0.33 | 0.34 | 0.36 | 0.31 | 0.3 | 0.33 | 0.33 |
| ROUGE-L-F1 | 0.29 | 0.31 | **0.32** | 0.29 | 0.28 | 0.21 | 0.24 | 0.26 | 0.27 | 0.23 |
| METEOR | 0.29 | 0.29 | **0.32** | 0.31 | 0.3 | 0.26 | 0.27 | 0.26 | 0.29 | 0.27 |
| CHRF | 38.51 | 37.77 | **40.8** | 40.05 | 39.34 | 34.3 | 37.86 | 36.27 | 38.61 | 37.96 |
| BERTScore-R | 0.88 | 0.88 | **0.89** | 0.88 | 0.88 | **0.89** | **0.89** | 0.88 | 0.88 | 0.88 |
| BERTScore-F1 | 0.88 | 0.88 | **0.89** | 0.88 | 0.88 | 0.87 | 0.88 | 0.88 | 0.88 | 0.87 |
| BARTScore | -3.65 | -3.69 | **-3.45** | -3.55 | -3.62 | -3.54 | -3.67 | -3.7 | -3.63 | -3.65 |
| GEval | 2.44 | 2.36 | 2.74 | **2.86** | 2.58 | 2.72 | 2.66 | 2.68 | 2.82 | 2.76 |
| AutoACU-R | 0.34 | 0.32 | 0.4 | **0.42** | 0.38 | 0.35 | 0.36 | 0.29 | 0.36 | 0.36 |
| AutoACU-F1 | 0.23 | 0.24 | 0.29 | **0.32** | 0.29 | 0.24 | 0.25 | 0.23 | 0.28 | 0.22 |
| Lite³Pyramid | 0.45 | 0.41 | **0.49** | 0.48 | 0.44 | 0.44 | 0.45 | 0.39 | 0.43 | **0.49** |
| AutoQAPyramid | 0.45 | 0.43 | 0.5 | **0.51** | 0.45 | 0.45 | 0.44 | 0.4 | 0.43 | 0.46 |
| nAutoQAPyramid | 0.41 | 0.41 | **0.47** | **0.47** | 0.41 | 0.33 | 0.4 | 0.38 | 0.39 | 0.39 |

Table 6: The metric scores of different summarization systems on 50 CNN/DM testing examples. The **bold** numbers are the best scores of each row.

# B   Annotation Details

## B.1   QA Generation Annotation

We collected human-written QA pairs on Amazon Mechanical Turk. QA pairs were collected for 500 reference summaries from CNN/DM. For each summary, we broke it into sentences. And for each sentence, we found all the predicates (verbs). There are 7.6 verbs per summary on average. So it's 3800 verbs in total. For each job (HIT), we presented the annotator with one verb highlighted in one sentence (Figure 6), and the annotator only needed to write QA pairs for this verb (on average 2.2 pairs per verb). It typically takes 1-2 minutes to finish one HIT. All 31 annotators we hired had gone through qualification tests and training. They knew the task pretty well. So, in total, it took about 3800 * 2 mins = 7600 minutes = 127 hours (about 4 hours per annotator) of working time. After QA pairs were written, we conducted a verification step to make sure QA pairs were generated correctly. Each QA pair was verified by two other annotators. In each HIT, one predicate (highlighted in one sentence) plus the QA pairs for this predicate were shown (Figure 7). Verification typically takes less than 1 minute to finish, i.e., 3800 * 2 annotators * 1 min = 7600 minutes = 127 hours.

## B.2   QA Presence Annotation

Same as QA generation, we collected QA presence labels on MTurk with 27 trained annotators. Presence labels were collected for 50 examples * 10 systems. And each judgment was provided independently by 3 annotators. In each HIT, we showed one system summary and the QA pairs for one predicate in the reference summary (Figure 8). It usually takes less than 2 minutes to finish one HIT. So, in total, 50 examples * 10 systems * 3 annotators * 7.6 predicates * 2 mins = 22800 minutes = 380 hours.

# C   Examples, Prompts, and Annotation UIs

We randomly picked summaries generated by fine-tuned models and ensured their QAPyramid scores were at least 0.2 higher than the corresponding ACU scores. We then analyzed the reasons why QAPyramid scores are higher than ACU scores.

First, QA pairs are finer-grained and thus allow partial credits. In Figure 3, the unit "British firm developed the vaccine" is not present in the summary but "What did something develop? it" is present because the summary says "took 30 years to develop". Similarly, in Example 2, "The Politician was played by Kate McKinnon" is not present but "Who did someone play? Politician" is present because the summary says "Show portrayed the former Secretary of State".

Second, our annotators made more lenient (based more on semantics than lexicon) judgments of "presence". because we instructed them that being present means the meaning of the QA pair is covered or implied by the summary or can be inferred from the summary, while the exact format/expression/wording can be different. In Figure 4, the ACU annotator labeled "Pep Guardiola has been linked with a switch" not present while labeling "Pep Guardiola has been linked to Manchester City" present probably because no explicit "switch" is mentioned in the summary. In contrast, our annotator labeled "What has someone been linked with? a switch to Manchester City" present because being linked with Manchester City implies a potential switch.

Third, since QA pairs are centered by predicates, when two predicates are close to each other, their QA pairs can have semantic overlap, which may cause repeated crediting for one piece of information. In Figure 5, the reference summary has "want to prevent" which contains two predicates: "want" and "prevent". When all QA pairs for "prevent" are labeled as present and thus credited, another QA "What does someone want? to prevent Vergara from destroying the embryos" gains one more credit.

**Reference**: Vaccine named RTS,S could be available by October, scientists believe . Will become the first approved vaccine for the world's deadliest disease . Designed for use in children in Africa, it can prevent up to half of cases . Experts hail 'extraordinary achievement' for British firm that developed it .

**Summary (generated by PEGASUS)**: The vaccine, known as RTS,S, took 30 years to develop but it is now hoped it can be used to save millions of lives. Scientists have worked on the vaccine for more than 20 years – at a cost of more than £330million. There is no licensed vaccine against malaria anywhere in the world. Researchers say they are hopeful the results will be sufficient for RTS,S to gain a licence from the EMA. The World Health Organisation could then recommend its use by October this year.

**ACU** = 0.09
Vaccine could be available by October, scientists believe. ✘
Vaccine is named RTS,S ✔
The vaccine will become the first approved vaccine for the world's deadliest disease ✘
The vaccine is for the world's deadliest disease ✘
The vaccine is designed for use in children ✘
The vaccine is designed for use in Africa ✘
The vaccine can prevent cases ✘
Up to half of cases can be prevented ✘
Experts hail 'extraordinary achievement' ✘
Experts hail 'extraordinary achievement' for British firm ✘
British firm developed the vaccine ✘

**QAPyramid** = 0.63
What could be available? Vaccine named RTS , S ✔
When could something be available? by October ✔
What is named something? Vaccine ✔
What is something named? RTS , S ✔
Who believes something? scientists ✔
What does someone believe? Vaccine named RTS , S could be available by October ✔
What will something become? the first approved vaccine for the world 's deadliest disease ✘
What will be approved? vaccine ✔
What was designed? it ✔
What was something designed for? use in children in Africa ✘
What can prevent something? it ✔
How many can something prevent? half of cases ✘
What does someone hail? extraordinary achievement ✘
Who hail something? Experts ✘
What developed something? British firm ✘
What did something develop? it ✔

Figure 3: An example of ACU and QAPyramid comparison.

**Reference**: The Bayern Munich boss has yet to commit his future to the German giants . Pep Guardiola has been linked with a switch to Manchester City . Former Bayern boss Ottmar Hitzfeld says Lucien Favre should replace him . Borussia Monchengladbach set for Champions League football next year .

**Summary (generated by BRIO-Ext)**: Bayern Munich manager Pep Guardiola has yet to commit his future to the Bundesliga giants. Guardiola, whose contract expires at the end of the 2015-16 season, has been linked with Manchester City. And former Bayern boss Ottmar Hitzfeld says Lucien Favre (left) should replace Guardiola at the Allianz Arena.

**ACU** = 0.43
The Bayern Munich boss has yet to commit his future ✔
his future is committed to the German giants ✘
Pep Guardiola has been linked with a switch ✘
Pep Guardiola has been linked to Manchester City ✔
Former Bayern boss Ottmar Hitzfeld says Lucien Favre should replace him ✔
Borussia Monchengladbach set for Champions League football ✘
Borussia Monchengladbach set next year ✘

**QAPyramid** = 0.82
What will someone commit? his future ✔
Who will commit something? The Bayern Munich boss ✔
Who will someone commit something to? the German giants ✔
Who has been linked with somthing? Pep Guardiola ✔
What has someone been linked with? a switch to Manchester City ✔
Who said something? Former Bayern boss Ottmar Hitzfeld ✔
What did someone say? Lucien Favre should replace him ✔
Who might be replaced? him ✔
Who might someone be replaced by? Lucien Favre ✔
Who is set for something? Borussia Monchengladbach ✘
What is someone set for? Champions League football ✘

Figure 4: An example of ACU and QAPyramid comparison.

**Reference**: Loeb says he filed the lawsuit and doesn't want want money from his "ex" Nick Loeb reportedly wants to prevent Vergara from destroying the embryos . Vergara spoke of freezing embryos with Loeb in a 2013 interview .

**Summary (generated by MatchSum)**: The 42-year-old actress and star of the hit TV sitcom "Modern Family" split from businessman Nick Loeb in May 2014. Loeb is suing the Colombian-born actress in Los Angeles to prevent Vergara from destroying their two embryos conceived through in vitro fertilization in November 2013, according to published reports by New York Daily News and In Touch magazine.

**ACU** = 0.33
Loeb filed the lawsuit. ✔
Loeb doesn't want money. ✘
Loeb doesn't want money from his ex. ✘
Nick Loeb wants to prevent Vergara. ✔
Nick Loeb is preventing from destroying the embryos. ✔
Vergara spoke of freezing embryos. ✘
Vergara spoke with Loeb. ✘
Vergara spoke in 2013. ✘
Vergara spoke in a 2013 interview. ✘

**QAPyramid** = 0.60
Who said something? Loeb ✘
What did someone say? he filed the lawsuit and does n't want want money from his " ex " ✘
Who filed something? Loeb ✔
What did someone file? the lawsuit ✔
Who doesn't want something? Loeb ✘
What doesn't someone want? money ✘
Who doesn't someone want something from? his " ex " ✘
Who wants something? Nick Loeb ✔
What does someone want? to prevent Vergara from destroying the embryos ✔
Who might prevent something? Nick Loeb ✔
What might someone prevent? destroying the embryos ✔
Who might someone prevent from doing something? Vergara ✔
Who may destroy something? Vergara ✔
What may be destroyed? the embryos ✔
Who spoke of something? Vergara ✘
What did someone speak of? freezing embryos with Loeb ✘
Where did someone speak of something? in a 2013 interview ✘
What did someone freeze? embryos ✔
Who froze something? Vergara ✔
Whom did someone freeze something with? Loeb ✔

Figure 5: An example of ACU and QAPyramid comparison.

| Method | Prompt |
|---|---|
| 1-shot Summary Generation | Article: Tomas Berdych set up a hotly-anticipated rematch [...] Summarize the above article in 3 sentences. Summary: Tomas Berdych beat Juan Monaco 6-3, 6-4 in the Miami Open last-eight [...] Article: {DOCUMENT} Summarize the above article in 3 sentences. Summary: |
| QA Generation | Read the following sentence. Produce question-answer pairs for the specified verb. You must give answer in a structured format: "Question: [your question] Answer: [your answer]", where [your question] and [your answer] is your generated question and answer, respectively. [Sentence] {SENTENCE} [Verb] {VERB} |
| QA Presence | Read the following summary. Then read a question and an answer. Answer whether the question and answer pair can be inferred from the summary. Please strictly output either [YES] or [NO]. [Summary] {SUMMARY} [Question] {QUESTION} [Answer] {ANSWER} |
| QA Presence Statement | Read the following summary. Then read a statement. Answer whether the statement pair can be inferred from the summary. Please strictly output either [YES] or [NO]. [Summary] {SUMMARY} [Statement] {STATEMENT} |
| QA to Statement | Convert the question and answer into a statement. Start your answer with "Statement:" Question: {QUESTION} Answer: {ANSWER} |

Table 7: LLM prompts used for generating and evaluating summaries.

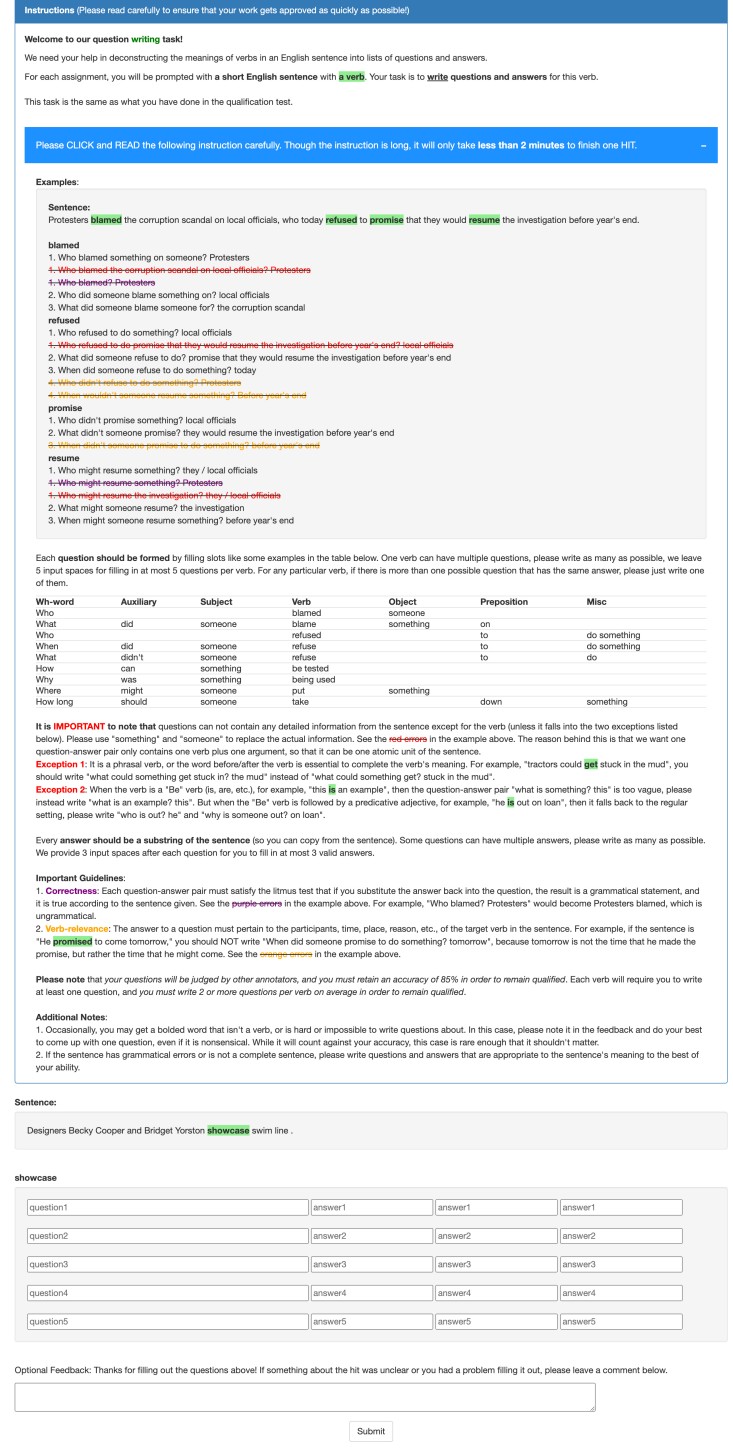

Figure 6: QA generation annotation instructions and UI.

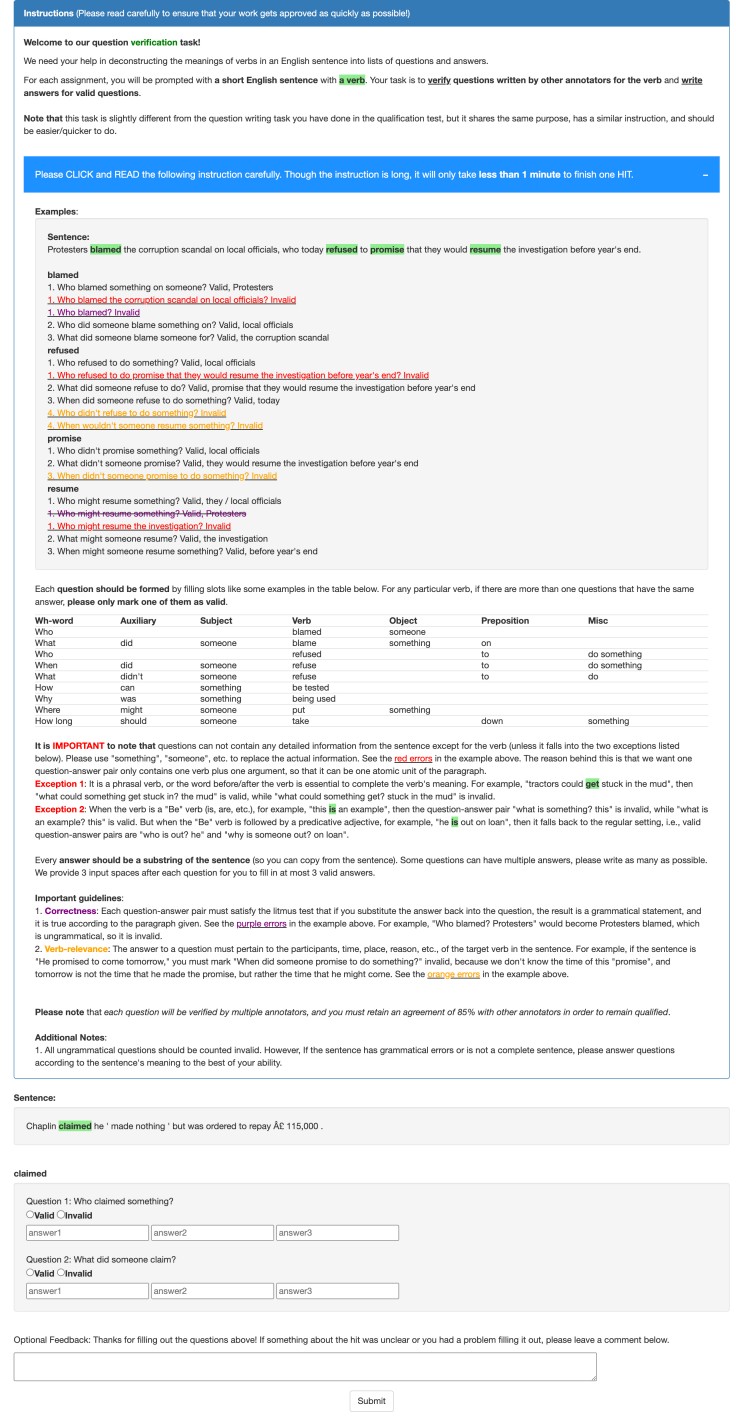

Figure 7: QA verification annotation instructions and UI.

Figure 8: QA presence judgment instructions and UI.

