# OpenReview forum: "QAPyramid: Fine-grained Evaluation of Content Selection for Text Summarization"
_colmweb.org/COLM/2025/Conference — COLM 2025_

### Official Review · Reviewer_1niJ · 2025-05-12

**Rating:** 7
**Confidence:** 2
**Ethics Flag:** 1

**Summary:**

This paper proposes a novel summarization metrics called QAPyramid. By grounding sub‐units in QA-SRL, each “Semantic Content Unit” becomes a well‐defined predicate–argument QA pair rather than the loosely specified SCUs or ACUs. This clarity improves inter‐annotator agreement and removes the need for expert‐only annotation.  In the summariation evaluation, the proposed metric can better capture the semantics of large language models, which are overly penalized by other metrics. With careful design, the paper systematically study the automated query-answer pair generation and detection using existing models. Through both semi-automated and fully-automated metric evaluation, the proposed metrics overal shows the most consistent high correlation with human annotations.

**Questions To Authors:**

1. During the annotation/QA detection, I can imagine answer paraphrasing happen a lot. Have you measured its sensitivity to paraphrasing in abstractive summarization, and if so, what strategies might mitigate any drop in recall?

**Reasons To Accept:**

1. The design and evaluation of QAPyramid covers both summarization and metric, it is very comprehensive and easy to follow.

2. QAPyramid’s automation is a major step toward making fine-grained summarization evaluation practical at scale. With a few domain-adaptation tweaks and an active‐learning guardrail, it could become a go-to metric for both researchers and practitioners.

3. Since LLM is widely used for summarization in real life, it amplify both the possibility and the stakes of automated summarization—and that makes evaluation more vital than ever.

**Reasons To Reject:**

1. The human end2end evaluation is relatively small (50 examples), which could hinder the truthfulness of the Table 1.

---

> ### Author Response · Authors · 2025-06-02
>
> > The human end2end evaluation is relatively small (50 examples), which could hinder the truthfulness of the Table 1.
>
> Yes, we acknowledge this limitation. Due to the budget limit, we weren’t able to collect a larger scale data. But we’d like to remind that though it’s 50 examples, it’s 50 examples * 10 systems which ended up with a large number of 8910 QA-level presence annotations (and each label was based on 3 independent annotations). Also, the QA pairs were collected for 500 examples (Line 144-145), which facilitates future work to conduct evaluations on a larger dataset.
>
> > During the annotation/QA detection, I can imagine answer paraphrasing happen a lot. Have you measured its sensitivity to paraphrasing in abstractive summarization, and if so, what strategies might mitigate any drop in recall?
>
> We have taken paraphrasing into consideration. As we mentioned in Line 182–184 and showed in the annotation UI in Figure 8, we instruct annotators to judge a QA pair as being present if its meaning is covered or implied by the system summary, or can be inferred from the system summary, while the exact format, expression, or wording can be different. We also see this instruction being enacted successfully. In Line 251-253, we mentioned that “our annotation guideline led to more lenient judgments of ‘being present’ based more on semantics than lexicon” which can be seen from an example in Figure 4 and the description in Line 830-838. Hence, we don’t think paraphrasing will lead to drop in QAPyramid scores. Similarly, when we automate QAPyramid, QA Presence detection is automated by few-shot GPT-4o. GPT-4o is considered to be good at handling paraphrasing.

---

> > ### Comment · Reviewer_1niJ · 2025-06-08
> >
> > Thanks for your response. It addressed my remaining concerns.

---

> > > ### Author Response · Authors · 2025-06-10
> > >
> > > We are happy to hear that our responses addressed your remaining concerns. Thank you!

---

> ### Author Response · Authors · 2025-06-06
>
> Dear reviewer, we are wondering if our responses addressed some of your concerns. Look forward to hearing back from you! Thanks!

---

### Official Review · Reviewer_u8Yo · 2025-05-12

**Rating:** 5
**Confidence:** 4
**Ethics Flag:** 1

**Summary:**

The paper introduces QAPyramid, a novel evaluation method for text summarization systems. It decomposes each reference summary into finer-grained question-answer (QA) pairs using the QA-SRL framework and assesses generated summaries based on the presence of these QA pairs. The authors constructed QA-SRL annotations from a subset of the CNN/DM dataset and applied the method to ten summarization models, comparing the resulting scores with those from other evaluation metrics.

**Questions To Authors:**

1. How many annotators were involved in Section 3.1? The context is unclear — Line 148 suggests a single annotator for QA rewriting, yet Line 169 states, "in 90.7% of cases, two annotators agree with each other." Could you clarify this discrepancy?

2. Line 173, Why not use three annotators with majority voting to enhance reliability? Have you conducted any preliminary annotation tasks to compare different settings?

3. It would be helpful to clearly distinguish between the appendix and the main body. Main findings should be presented in the main body, with the appendix reserved for additional details. For instance, in line 277, the text states, “We report the results in Table 3,” but it should emphasize that Table 3 is located in the appendix and serves as supplementary information.

**Reasons To Accept:**

Designing effective evaluation metrics is vital for fairly assessing summarization models and can significantly contribute to the research community.

**Reasons To Reject:**

1. The proposed evaluation method emphasizes predicates while overlooking other important aspects of summary quality. Additionally, it primarily focuses on recall, potentially limiting its effectiveness in assessing the precision of system-generated summaries.

2. The paper assesses metrics against QAPyramid human annotations, assuming higher accuracy. However, it would be beneficial to provide empirical evidence supporting this claim. Given the inter-annotator agreement results in the AQ presence detection evaluation, the purported superiority of QAPyramid annotations over existing methods is not clearly demonstrated.

3. The experimental analysis could be further developed to provide more substantive insights. For instance, the takeaways from Section 3.4 are somewhat unclear, as the findings align with existing metric trends in Table 1 without offering distinct or noteworthy observations. Additionally, higher scores for a metric compared to another do not necessarily indicate superior effectiveness.

---

> ### Author Response · Authors · 2025-06-02
>
> > The proposed evaluation method emphasizes predicates while overlooking other important aspects of summary quality.
>
> Would you mind elaborating on the “other important aspects” you are referring to?
>
> QA-SRL is centered around predicates (verbs), but it also has arguments. Predicate-argument structure is a key concept in semantics that breaks down sentences into core components. In other words, the meaning of a sentence is conveyed by the predicate-argument structure (see Line 108-119). Nonetheless, we acknowledge that not all predicate types are covered by QA-SRL, e.g., deverbal nominalizations (Klein et al., 2020), discourse relations (Pyatkin et al., 2020), and adjectival semantics (Pesahov et al., 2023). We will discuss our limitations in the extra page of the final version.
>
> Klein et al., 2020: [QANom: Question-Answer driven SRL for Nominalizations](https://aclanthology.org/2020.coling-main.274/)
>
> Pyatkin et al., 2020: [QADiscourse - Discourse Relations as QA Pairs: Representation, Crowdsourcing and Baselines](https://aclanthology.org/2020.emnlp-main.224/)
>
> Pesahov et al., 2023: [QA-Adj: Adding Adjectives to QA-based Semantics](https://dmr2023.github.io/pdf/2023.dmr-1.8.pdf)
>
>
> > Additionally, it primarily focuses on recall, potentially limiting its effectiveness in assessing the precision of system-generated summaries.
>
> Yes, QAPyramid is a recall-based metric, same as the original Pyramid method and its variants e.g., ACU. Nonetheless, we also introduce normalized QAPyramid (nQAPyramid) (Line 227-228), which penalizes long and repetitive summaries. It takes precision into consideration, e.g., if a summary has high recall while being shorter than the reference summary, it means it also has high precision.
>
> > The paper assesses metrics against QAPyramid human annotations, assuming higher accuracy. However, it would be beneficial to provide empirical evidence supporting this claim. Given the inter-annotator agreement results in the AQ presence detection evaluation, the purported superiority of QAPyramid annotations over existing methods is not clearly demonstrated.
>
> We demonstrated the superiority of QAPyramid from 3 angles:
> 1. QAPyramid achieved high/competitve inter-annotator agreement (0.74 Krippendorff’s alpha) for QA presence judgment, **compared to other Pyramid-style human evaluations** (see Line 200-201): 0.75 Liu et al. (2023b), 0.73 Zhang & Bansal (2021), and 0.66 Bhandari et al. (2020). And if we **compare to non-Pyramid-style human evaluation**, SummEval (Fabbri et al., 2021) reported 0.49 Krippendorff’s alpha for crowd-sourced workers and 0.71 for experts. Note that we used crowd-sourced workers.
> 2. QAPyramid is more fine-grained. Each QA pair corresponds to a “minimal” predicate-argument relation. This is shown in Figure 1 and Line 245-249.
> 3. QAPyramid is more systematic (Line 114-119). It’s based on the well-established QA-SRL framework. Therefore, the QA generation step is less ambiguous and thus more reproducible, which is demonstrated by the high agreement for QA generation (Line 169-170): “we find a high inter-annotator agreement: in 90.7% of cases, two annotators agree with each other, and in 89.7% of cases, the question is labeled as valid by both annotators.” Not to mention that QA-SRL is crowdsourable (FitzGerald et al., 2018), which makes it more scalable.
>
> Fabbri et al., 2021: [SummEval: Re-evaluating Summarization Evaluation](https://arxiv.org/abs/2007.12626)
>
> Liu et al. 2023b: [Revisiting the Gold Standard: Grounding Summarization Evaluation with Robust Human Evaluation](https://aclanthology.org/2023.acl-long.228/)
>
> Zhang & Bansal, 2021: [Finding a Balanced Degree of Automation for Summary Evaluation](https://arxiv.org/abs/2109.11503)
>
> Bhandari et al. 2020: [Re-evaluating Evaluation in Text Summarization](https://aclanthology.org/2020.emnlp-main.751/)
>
> FitzGerald et al., 2018: [Large-Scale QA-SRL Parsing](https://arxiv.org/abs/1805.05377)

---

> > ### Author Response · Authors · 2025-06-02
> >
> > > The experimental analysis could be further developed to provide more substantive insights. For instance, the takeaways from Section 3.4 are somewhat unclear, as the findings align with existing metric trends in Table 1 without offering distinct or noteworthy observations. Additionally, higher scores for a metric compared to another do not necessarily indicate superior effectiveness.
> >
> > It is correct that QAPyramid has a similar system ranking compared to existing metrics like ACU and ROUGE. They all rank BRIO as the top system. But this does not necessarily mean QAPyramid offers no distinct or noteworthy observation.
> >
> > First of all, this alignment on system level ranking verifies that QAPyramid is doing the right thing. It would be questionable if QAPyramid suddenly ranked BRIO as the worst system.
> >
> > Second, system-level ranking is an aggregated signal. If you look at the summary level (e.g., Figure 1), QAPyramid provides better and more fine-grained interpretability compared to ROUGE and ACU. It clearly shows which pieces of information are missing from the system summary.
> >
> > Lastly, yes QAPyramid indeed gives higher scores compared to other metrics. We do not claim higher scores means superior effectiveness. We’d like to understand why it gives higher scores in Line 245-254. We found (1) since QAPyramid is more fine-grained, it gives credits to partial correctness which leads to higher scores (2) our annotation guideline led to more lenient judgments of “being present” based more on semantics than lexicon (3) the predicate-centered nature of QA-SRL may cause one piece of information to be credited multiple times. All these contribute to higher scores. But we do not claim QAPyramid is better simply because it gives higher scores.
> > We will make this clear in the final version of the paper.
> >
> > > How many annotators were involved in Section 3.1? The context is unclear — Line 148 suggests a single annotator for QA rewriting, yet Line 169 states, "in 90.7% of cases, two annotators agree with each other." Could you clarify this discrepancy?
> >
> > A single annotator for QA rewriting is correct. However, as mentioned in Line 158, “For each of the collected QA pairs, following the original QA-SRL work (FitzGerald et al. 2018), we ask two other annotators to verify it.” This is why we have “in 90.7% of cases, two annotators agree with each other”, i.e., two verifiers agree with each other in 90.7% of time. Also, “in 89.7% of cases, the question is labeled as valid by both annotators” means the two verifiers agree with the writer in 89.7% of time. We will clarify this in the final version of the paper.
> >
> > > Line 173, Why not use three annotators with majority voting to enhance reliability? Have you conducted any preliminary annotation tasks to compare different settings?
> >
> > It can be seen as 3 annotators: one writer + two verifiers. And “in the end, we only keep QA pairs that are verified to be valid by both annotators”, meaning we only keep QA pairs that 3 annotators agree with each other. For QA presence detection (Section 3.2), we collect labels from 3 annotators and take the majority vote (Line 198).
> >
> > We adopted the annotation setup (e.g, 1 writer + 2 verifiers) from FitzGerald et al. (2018), the original large-scale QA-SRL data collection work, which validated the effectiveness of this data annotation process. We will clarify this in the next draft.
> >
> > > It would be helpful to clearly distinguish between the appendix and the main body. Main findings should be presented in the main body, with the appendix reserved for additional details. For instance, in line 277, the text states, “We report the results in Table 3,” but it should emphasize that Table 3 is located in the appendix and serves as supplementary information.
> >
> > We will bring Table 3 back to the main using the extra page of the final version of this paper.

---

> > > ### Comment · Reviewer_u8Yo · 2025-06-09
> > >
> > > Thank you to the authors for the detailed response. After reviewing the rebuttal and considering the insights from other reviewers, I have decide to increase my scores.

---

> > > > ### Author Response · Authors · 2025-06-10
> > > >
> > > > Thank you for increasing your score! We are happy to see our responses address some of your concerns. In the meantime, it would be great if you could let us know your remaining concerns of accepting this work. It would help us to improve our work in the next version. Thank you a lot!

---

> ### Author Response · Authors · 2025-06-06
>
> Dear reviewer, we are wondering if our responses addressed some of your concerns. Look forward to hearing back from you! Thanks!

---

### Official Review · Reviewer_YNdU · 2025-05-15

**Rating:** 6
**Confidence:** 3
**Ethics Flag:** 1

**Summary:**

The paper introduces QAPyramid, a human-evaluation protocol that decomposes each reference summary into fine-grained question–answer pairs produced with the QA-SRL framework, allowing partial-credit reasoning about content selection. By crowdsourcing 8.6 k QA pairs for 500 CNN/DM references and 8.9 k presence judgements over ten summarization systems, the authors show that QAPyramid is more granular and reproducible than Pyramid/ACU and that its semi- and fully-automatic versions correlate better with human judgements than existing metrics.

**Questions To Authors:**

Can you discuss how your work relates to the findings in the following paper? It seems like they also break down the annotation process to achieve higher inter-annotator agreement.
 Pagnoni, Artidoro, Vidhisha Balachandran, and Yulia Tsvetkov. "Understanding factuality in abstractive summarization with FRANK: A benchmark for factuality metrics." https://arxiv.org/abs/2104.13346

**Reasons To Accept:**

1. QAPyramid is more fine grained and precise than current evaluation approaches even those based on pyramid/ACUs/SCUs.
2. QA SRL can be annotated with high quality via crowdsourcing making it more practically useful and accessible compared to expert-annotated SCUs and ACUs.
3. SemiAutoQAPyramid and AutoQAPyramid achieve the highest Kendall correlations against gold scores among the metrics tested. However, these are also more targeted metrics compared to the other metrics.

**Reasons To Reject:**

1. There is no significant demonstration that QA Pyramid is a good approach for assessing the performance of summarization systems. It would be useful to compare QA Pyramid to some other evaluation that is accepted. For example comparing it to human judgement of the quality of the summaries. Without this comparison it is hard to know whether systems with higher QA Pyramid scores are actually better systems.
2. While some agreement is measured, it is not a standard metric for agreement and is not compared to other approaches in the literature as a reference. Additionally, the agreement is only reported on the verification task rather than the annotation task.
3. The penalty extension to handle repetitive content seems unnecessary. Specifically, this could be handled by improved decoding techniques where the repeated content is removed.
4. Issues with synonyms: how do annotators know that designing a vaccine is the same as developing a vaccine? How can annotators know what constitutes a match?
5. Issues with coreference: it refers to the vaccine, but without context it could have been a different entity. This problem could be solved by resolving the coreference.

---

> ### Author Response · Authors · 2025-06-02
>
> > There is no significant demonstration that QA Pyramid is a good approach for assessing the performance of summarization systems. It would be useful to compare QA Pyramid to some other evaluation that is accepted. For example comparing it to human judgement of the quality of the summaries. Without this comparison, it is hard to know whether systems with higher QA Pyramid scores are actually better systems.
>
> We’d like to point out that QAPyramid is a new human evaluation method. Although many other human evaluations have been used in previous works (e.g., direct rating, see our discussions in Section 2), we want to show that QAPyramid is a better and more reliable way to conduct human evaluation.
>
> Usually, a human evaluation approach is considered reliable **if it achieves high inter-annotator agreement**.
>
> We demonstrated that QAPyramid is a good human evaluation approach from 3 angles:
> 1. QAPyramid achieved high/competitive inter-annotator agreement (0.74 Krippendorff’s alpha) for QA presence judgment, **compared to other Pyramid-style human evaluations** (see Line 200-201): 0.75 Liu et al. (2023b), 0.73 Zhang & Bansal (2021), and 0.66 Bhandari et al. (2020). And if we **compare to non-Pyramid-style human evaluation**, SummEval (Fabbri et al., 2021) reported 0.49 Krippendorff’s alpha for crowd-sourced workers and 0.71 for experts. Note that we used crowd-sourced workers.
> 2. QAPyramid is more fine-grained. Each QA pair corresponds to a “minimal” predicate-argument relation. This is shown in Figure 1 and Line 245-249.
> 3. QAPyramid is more systematic. It’s based on the well-established QA-SRL framework (Line 114-119). Therefore, the QA generation step is less ambiguous and thus more reproducible, which is demonstrated by the high agreement for QA generation (Line 169-170): “we find a high inter-annotator agreement: in 90.7% of cases, two annotators agree with each other, and in 89.7% of cases, the question is labeled as valid by both annotators.” Not to mention that QA-SRL is crowdsourable (FitzGerald et al., 2018), which makes it more scalable.
>
> Fabbri et al., 2021: [SummEval: Re-evaluating Summarization Evaluation](https://arxiv.org/abs/2007.12626)
>
> Liu et al. 2023b: [Revisiting the Gold Standard: Grounding Summarization Evaluation with Robust Human Evaluation](https://aclanthology.org/2023.acl-long.228/)
>
> Zhang & Bansal, 2021: [Finding a Balanced Degree of Automation for Summary Evaluation](https://arxiv.org/abs/2109.11503)
>
> Bhandari et al. 2020: [Re-evaluating Evaluation in Text Summarization](https://aclanthology.org/2020.emnlp-main.751/)
>
> FitzGerald et al., 2018: [Large-Scale QA-SRL Parsing](https://arxiv.org/abs/1805.05377)
>
> > While some agreement is measured, it is not a standard metric for agreement and is not compared to other approaches in the literature as a reference.
>
> We report agreement in two places.
>
> First, for QA generation, one annotator wrote the QA pairs and then two other annotators reviewed/verified them. We reported the agreement of the two verifiers in Line 169-170: “in 90.7% of cases, two annotators agree with each other.” In this case, we have only two verifiers for each example, so they either agree or disagree. The original QA-SRL work (FitzGerald et al., 2018) also reported a similar agreement rate of 90.9%.
>
> Second, for QA presence, the presence of each QA pair is judged by 3 annotators. In this case, we computed the widely used Krippendorff’s alpha and compared it to previous works in Line 197-201: “The average inter-annotator agreement (IAA) is 0.74 (Krippendorff’s alpha). For reference, Liu et al. (2023b) reported an IAA of 0.75, Zhang & Bansal (2021) reported 0.73, and Bhandari et al. (2020) reported 0.66.”
>
> > Additionally, the agreement is only reported on the verification task rather than the annotation task.
>
> We mentioned in Line 170 that “in 89.7% of cases, the question is labeled as valid by both annotators”, which means in 89.7% of cases the two verifiers agree with each other and in the meantime agree with the QA writer.
>
> Using one writer and two verifiers is a setup we adopted from the original QA-SRL work (FitzGerald et al., 2018). They proved this setup to be effective for collecting large-scale QA-SRL data.
>
> > The penalty extension to handle repetitive content seems unnecessary. Specifically, this could be handled by improved decoding techniques where the repeated content is removed.
>
> The repetition penalty is 1 (no penalty) if the summary has no repetition. Yes, changing decoding techniques can avoid repeated text. However, it is not guaranteed that we will never get repetitive text. We present QAPyramid as a generic human evaluation protocol that should handle any kind of model generation under any decoding techniques. This repetition penalty is ensuring that generations that are not decoded with good hyperparameters will also get correct judgments.

---

> > ### Author Response · Authors · 2025-06-02
> >
> > > Issues with synonyms: how do annotators know that designing a vaccine is the same as developing a vaccine? How can annotators know what constitutes a match?
> >
> > As we mentioned in Line 182-184, “we instruct annotators to judge a QA pair as being present if its meaning is covered or implied by the system summary, or can be inferred from the system summary, while the exact format, expression, or wording can be different.” We also see this instruction being enacted successfully. In Line 251-253, we mentioned that “our annotation guideline led to more lenient judgments of ‘being present’ based more on semantics than lexicon” which can be seen from an example in Figure 4 and the description in Line 830-838.
> >
> > When annotators judge the presence of each QA pair, they will also see the reference summary to gain necessary context information. The annotation UI is shown in Figure 8, which has detailed instructions. All the annotators we hired had gone through a qualification test (Line 187-188), so they are capable of judging QA presence.
> >
> > Such semantic judgements, while seeing the full context, are the standard in decomposition-based methods, like the original Pyramid  (Nenkova & Passonneau, 2004) and ACU (Liu et al. 2023b). It basically corresponds to entailment judgements.
> >
> > Nenkova & Passonneau, 2004: [Evaluating Content Selection in Summarization: The Pyramid Method](https://aclanthology.org/N04-1019/)
> >
> > > Can you discuss how your work relates to the findings in the following paper? It seems like they also break down the annotation process to achieve higher inter-annotator agreement. Pagnoni, Artidoro, Vidhisha Balachandran, and Yulia Tsvetkov. "Understanding factuality in abstractive summarization with FRANK: A benchmark for factuality metrics." https://arxiv.org/abs/2104.13346
> >
> > First of all, FRANK is a benchmark for factuality evaluation. Factuality is to evaluate whether the system summary is semantically consistent with the source document. In contrast, our approach is to evaluate content selection (or you can roughly call it ``quality’’) of the summary, i.e., whether the summary selects the right content to include. **Therefore, we evaluate the system summary against the reference summary (not the source document).**
> >
> > Second, FRANK breaks down the annotation process in the sense of introducing a typology of factual errors and collecting error annotations for each summary sentence.In other words, FRANK breaks the definition of "factuality" down into different error categories and thus allow annotators to more accurately describe what problem they see with the summary. In contrast, **we have a completely different type of breakdown**. We break down each reference summary into smaller semantic units (QAs) and then check if the system summary entails each QA, which allows more fine-grained and more comprehensive judgement.
> >
> > We discussed how our work relates to other similar/related works in Section 2, especially the *Human Evaluation for Text Summarization* and *Automatic Evaluation for Text Summarization* paragraphs.

---

> > > ### Comment · Reviewer_YNdU · 2025-06-06
> > >
> > > Thank you to the authors for their clarifications. Overall, the authors seem to support their choices and disagree with most of the feedback I provided. For example:
> > >
> > > > Usually, a human evaluation approach is considered reliable if it achieves high inter-annotator agreement.
> > >
> > > Agreement cannot be the only measure of quality for human evaluation. We can achieve perfect agreement on tasks that don't evaluate anything meaningful. So I stand by the feedback that this evaluation approach would benefit from additional validation or empirical demonstrations of its advantages.

---

> > ### Author Response · Authors · 2025-06-06
> >
> > Thanks for your reply! We appreciate your feedback, but we are a bit confused by your response. We were mainly trying to provide clarification in our previous response, and we didn’t disagree with most of your feedback. Instead, we are happy to accept any constructive feedback you give! The quality of QAPyramid was justified not only by agreement but also by being more fine-grained and systematic. Would you mind elaborating on which part you think we disagree with you and what "additional validation or empirical demonstrations" you think we should include?

---

> > > ### Author Response · Authors · 2025-06-10
> > >
> > > Dear reviewer, as the rebuttal period is approaching its end, we wanted to send a quick reminder here in case you missed our previous response. We are looking forward to hear back from you. Thank you!

---

### Official Review · Reviewer_tvdb · 2025-05-18

**Rating:** 6
**Confidence:** 3
**Ethics Flag:** 1

**Summary:**

This paper describes a method for human evaluation for content selection in the text summarization task. The method, called QAPyramid, enables fine-grained evaluation for the task that shows high inter-annotator agreement. The paper also presents an LLM-based automatic evaluation pipeline that shows high correlation with human judgements on the task.

The paper conducts a series of experiments on 10 summarization models, both fine-tuned and 1-shot prompted, to demonstrate the utility of the proposed evaluation. The proposed method for evaluation is interesting and seems to provide better insights into performance than existing methods. However, some details, such as the application of the repetition penalty and length penalty are unclear – the calculation and explanation of these seem to be rather opaque, making their significance rather unclear in the paper. The experiments are conducted with 50 examples, which is a relatively small dataset. While I understand budget constraints here, I wonder if the conclusions drawn from a small sample like this are generalizable – especially since the sample is from a single dataset (CNN / Daily Mail).

**Questions To Authors:**

* How long did manual generation of QA take?
* How long did human judgement of QA presence take?
* Were the 50 examples for the subsample chosen randomly? Is there any distribution of the domains / use-cases it covers?

Minor typos:
* Line 131: systermatical → systematic
* Line 81: off-the-shell → off-the-shelf

**Reasons To Accept:**

* The proposed method for summarization evaluation is interesting and novel – it reduces inter-annotator disagreement and provides more fine-grained insights into content selection than existing methods.
* The experiments cover a variety of models, both fine-tuned and LLM-based.
* The automatic metrics for QA generation and presence detection are useful to scale up the method to be applicable to a various datasets without human annotation or evaluation.

**Reasons To Reject:**

* My primary concern with this method is about its scalability and generalizability – the experiments are based on 50 examples from the CNN / DailyMail. I wonder if the method will be more broadly applicable and how it will affect summarization evaluation as LLMs have a broad range of summarization tasks, domains, and languages as possible applications.
* Would qualitative analysis or application of the automatic algorithms to other summarization datasets be beneficial here to understand the generalizability?

---

> ### Author Response · Authors · 2025-06-02
>
> > My primary concern with this method is about its scalability and generalizability – the experiments are based on 50 examples from the CNN / DailyMail. I wonder if the method will be more broadly applicable and how it will affect summarization evaluation as LLMs have a broad range of summarization tasks, domains, and languages as possible applications.
>
> Due to the budget limit, we weren’t able to collect a large number of examples from multiple datasets. Note that even just with 50 examples * 10 systems, we ended up collecting a large number of 8.9K QA presence labels (and each label was based on 3 independent annotations). Also, QA pairs were collected for 500 examples from CNN/DM (Line 144-145).
>
> Nonetheless, by design, QAPyramid can be applied to other tasks, domains, and languages. For any summarization task that has reference summaries, people can break reference summaries into QA pairs following the QA-SRL schema, and then check the presence of each QA pair against the system summary. We release our data, our data annotation harness, AutoQAPyramid calculation scripts to support future work. Hence, we foresee this evaluation paradigm to be generalizable.
>
> > Would qualitative analysis or application of the automatic algorithms to other summarization datasets be beneficial here to understand the generalizability?
>
> QAPyramid is a human evaluation protocol, which we suggest is more reliable than other ways of collecting human judgments. AutoQAPyramid is a way to automate QAPyramid, and we demonstrate it to be an effective automation by the correlation results (Table 2).
>
> In the next version, we will apply AutoQAPyramid to other summarization evaluation benchmarks that have existing human judgments collected. Concretely, we will run our automatic metric on the standard automatic benchmark of SummEval (Fabbri et al., 2021). We will show the correlations between AutoQAPyramid and human judgments from SummEval, despite that their human judgments were collected following a different protocol. This will help demonstrate the generalizability of our method.
>
> Fabbri et al., 2021: [SummEval: Re-evaluating Summarization Evaluation](https://arxiv.org/abs/2007.12626)
>
> > How long did manual generation of QA take?
>
> TLDR:
> The manual generation took about 254 hours in total, which is about 8 hours per annotator for 31 annotators. We will include these statistics in the Appendix in the next version.
>
> Details:
> We collected human-written QA pairs on Amazon Mechanical Turk. QA pairs were collected for 500 reference summaries from CNN/DM. For each summary, we broke it into sentences. And for each sentence, we found all the predicates (verbs). There are 7.6 verbs per summary on average. So it’s 3800 verbs in total. For each job (HIT), we presented the annotator with one verb highlighted in one sentence (Figure 6), and the annotator only needed to write QA pairs for this verb (on average 2.2 pairs per verb). It typically takes 1-2 minutes to finish one HIT. All 31 annotators we hired had gone through qualification tests and training. They knew the task pretty well. So, in total, it took about 3800 * 2 mins = 7600 minutes = 127 hours (about 4 hours per annotator) of working time. After QA pairs were written, we conducted a verification step to make sure QA pairs were generated correctly. Each QA pair was verified by two other annotators. In each HIT, one predicate (highlighted in one sentence) plus the QA pairs for this predicate were shown (Figure 7). Verification typically takes less than 1 minute to finish, i.e., 3800 * 2 annotators * 1 min = 7600 minutes = 127 hours.
>
>
> > How long did human judgement of QA presence take?
>
> TLDR:
> The presence detection took about 380 hours in total, which is about 14 hours per annotator for 27 annotators. We will include these statistics in the Appendix in the next version.
>
> Details:
> Same as QA generation, we collected QA presence labels on MTurk with 27 trained annotators. Presence labels were collected for 50 examples * 10 systems. And each judgment was provided independently by 3 annotators. In each HIT, we showed one system summary and the QA pairs for one predicate in the reference summary (Figure 8). It usually takes less than 2 minutes to finish one HIT.  So, in total, 50 examples * 10 systems * 3 annotators * 7.6 predicates * 2 mins = 22800 minutes = 380 hours.
>
> > Were the 50 examples for the subsample chosen randomly?
>
> Yes. We will clarify this in the next version.

---

> > ### Author Response · Authors · 2025-06-02
> >
> > > However, some details, such as the application of the repetition penalty and length penalty are unclear – the calculation and explanation of these seem to be rather opaque, making their significance rather unclear in the paper.
> >
> > Would you mind specifying which part is opaque? We will try to explain them as follows. But it would be more helpful if you can specify the part you find opaque. Thanks!
> >
> > As we described in Line 206-211, length penalty was introduced by the ACU method (Liu et al. 2023b published at ACL 2023). We directly use their original formula: $p^l_i = e^{min(0, \frac{1-\frac{|s_i|}{|r_i|}}{\alpha})}$. It scales the original ACU score down for summaries that are longer than the reference summary.
> >
> > $|s_i|$ is the length of the system summary, and $|r_i|$ is the length of reference summary. For example, if $|s_i| = 100$, $|r_i|=50$, and $\alpha=6$, then this penalty value is $e^{min(0, \frac{1-\frac{100}{50}}{6})} = 0.85$. So if the original ACU is 0.7, nACU will be 0.7*0.85 = 0.6. ACU (or QAPyramid) score is recall-based, i.e., more information in the summary leads to higher scores. In an extreme case where the summary copies the whole document, ACU (or QAPyramid) score will be 100% (Line 204-206). Usually a longer summary contains more information. This penalty is to discount the ACU (or QAPyramid) score based on length, as a type of length normalization.
> >
> > The hyperparameter $\alpha$ is used to control the “strength” of this penalty. For example, if $\alpha$ is very large (towards positive infinity), this penalty value will always be close to 1, i.e., no penalty. In practice, $\alpha$ is set by decorrelating nACU with summary length (Line 210 - 211).
> >
> > However, this length penalty has one drawback. It assumes that longer length means more information. But it’s not always true. We found that, sometimes, few-shot LLM-generated summaries get stuck into a repetition loop due to the degeneration problem (Holtzman et al., 2020). An example is given in Figure 2. In this case, the longer length does not include more information, i.e., it won’t lead to higher ACU or QAPyramid scores. So it does not make sense to set $\alpha$ by decorrelating nACU with summary length.
> >
> > To deal with this repetition issue, we introduced the repetition penalty (Line 220-222 and Appendix A.2), $p^r_i = 1-rp_i$. $rp_i$ is the repetition rate of summary $s_i$ . For example, if the summary is *“Gwyneth Paltrow and Chris Martin are seen together with their children in a family photo . Gwyneth Paltrow and Chris Martin are seen together with their children in a family photo . Gwyneth Paltrow and Chris Martin are seen together with their children in a family photo .”*  The actual content is just *“Gwyneth Paltrow and Chris Martin are seen together with their children in a family photo .”* The repetition rate is $\frac{2}{3}$. And the effective length (length without repetition) is $\frac{1}{3}$ * summary length, i.e., the length of *“Gwyneth Paltrow and Chris Martin are seen together with their children in a family photo .”*
> >
> > Putting these two together, we normalize QAPyramid with both penalties: nQAPyramid = $p^r_i∗ p^l_i$ ∗ QAPyramid.
> >
> > The actual implementation of these two penalties is straightforward and quick to run. Also, we don’t view them as significant contributions. They are just simple fixes on top of QAPyramid to penalize long length and repetition.
> >
> > Liu et al. 2023b: [Revisiting the Gold Standard: Grounding Summarization Evaluation with Robust Human Evaluation](https://aclanthology.org/2023.acl-long.228/)
> >
> > Holtzman et al., 2020: [The Curious Case of Neural Text Degeneration](https://arxiv.org/abs/1904.09751)

---

> > ### Comment · Reviewer_tvdb · 2025-06-10
> >
> > Thank you for the discussion and response to my questions. I have edited the score in my review based on the author response.
> >
> > I still have some concerns on the overall generalizability of the method to other domains (and languages) and the effort of manual annotation / generation. It would be great to mention these limitations in the next version of the paper.

---

> > > ### Author Response · Authors · 2025-06-10
> > >
> > > Thank you for increasing your score! We will be certain to discuss the limitations in the next version of our paper.

---

> ### Author Response · Authors · 2025-06-06
>
> Dear reviewer, we are wondering if our responses addressed some of your concerns. Look forward to hearing back from you! Thanks!

---

> ### Author Response · Authors · 2025-06-10
>
> Dear reviewer, as the rebuttal period is approaching its end, we would like to send a friendly reminder regarding our response to ensure that you have had an opportunity to review it. We hope that our response will allow you to revisit your score. Thank you very much!

---

### Decision · Program_Chairs · 2025-07-08

**Decision:**

Accept

**Comment:**

This paper proposes a method to systematise the pyramid evaluation protocol. There was some good discussion between the reviewers and the authors regarding how to evaluate summarisation evaluation methods. The premise of the paper is that if we accept that Pyramid is a good summarisation evaluation, then QAPyramid and its automated version are  more consistent and fine-grained ways to do it. This is well-supported in the experiments. If accepted, I would really like the authors to deliver on this promise:

''In the next version, we will apply AutoQAPyramid to other summarization evaluation benchmarks that have existing human judgments collected. Concretely, we will run our automatic metric on the standard automatic benchmark of SummEval (Fabbri et al., 2021). We will show the correlations between AutoQAPyramid and human judgments from SummEval, despite that their human judgments were collected following a different protocol.''